# MK2a inhibitor CMPD1 abrogates chikungunya virus infection by modulating actin remodeling pathway

**Prabhudutta Mamidi**[1], **Tapas Kumar Nayak**[2,3], **Abhishek Kumar**[1,4], **Sameer Kumar**[1,5], **Sanchari Chatterjee**[1,6], **Saikat De**[1,6], **Ankita Datey**[1,7], **Soumyajit Ghosh**[1,6], **Supriya Suman Keshry**[1,7], **Sharad Singh**[1,7], **Eshna Laha**[1,6], **Amrita Ray**[1,6], **Subhasis Chattopadhyay**[2], **Soma Chattopadhyay**[1] *

1 Institute of Life Sciences, Bhubaneswar, India, 2 National Institute of Science Education and Research, Bhubaneswar, India, 3 Center for Translational Medicine, Lewis Katz School of Medicine, Temple University, Philadelphia, Pennsylvania, United States of America, 4 Department of Oral Biology, University of Florida College of Dentistry, Gainesville, Florida, United States of America, 5 Department of Microbiology and Immunology, Carver College of Medicine, University of Iowa, Iowa City, Iowa, United States of America, 6 Regional Centre for Biotechnology, Faridabad, India, 7 KIIT school of Biotechnology, Bhubaneswar, India

* sochat.ils@gmail.com

**Data Availability Statement:** All microarray data are available from the Array express database (accession number E-MTAB-6645).

## Abstract

Chikungunya virus (CHIKV) epidemics around the world have created public health concern with the unavailability of effective drugs and vaccines. This emphasizes the need for molecular understanding of host-virus interactions for developing effective targeted antivirals. Microarray analysis was carried out using CHIKV strain (Prototype and Indian) infected Vero cells and two host isozymes, MAPK activated protein kinase 2 (MK2) and MAPK activated protein kinase 3 (MK3) were selected for further analysis. The substrate spectrum of both enzymes is indistinguishable and covers proteins involved in cytokines production, endocytosis, reorganization of the cytoskeleton, cell migration, cell cycle control, chromatin remodeling and transcriptional regulation. Gene silencing and drug treatment were performed *in vitro* and *in vivo* to unravel the role of MK2/MK3 in CHIKV infection. Gene silencing of MK2 and MK3 abrogated around 58% CHIKV progeny release from the host cell and a MK2 activation inhibitor (CMPD1) treatment demonstrated 68% inhibition of viral infection suggesting a major role of MAPKAPKs during late CHIKV infection *in vitro*. Further, it was observed that the inhibition in viral infection is primarily due to the abrogation of lamellipodium formation through modulation of factors involved in the actin cytoskeleton remodeling pathway. Moreover, CHIKV-infected C57BL/6 mice demonstrated reduction in the viral copy number, lessened disease score and better survivability after CMPD1 treatment. In addition, reduction in expression of key pro-inflammatory mediators such as CXCL13, RAGE, FGF, MMP9 and increase in HGF (a CHIKV infection recovery marker) was observed indicating the effectiveness of the drug against CHIKV. Taken together it can be proposed that MK2 and MK3 are crucial host factors for CHIKV infection and can be considered as important target for developing effective anti-CHIKV strategies.

**Funding:** This study has been funded by the Department of Science and Technology (DST-SERB), New Delhi, India, vide grant no EMR/2016/000854. https://www.serbonline.in/SERB/HomePage. It was also supported by Institute of Life Sciences, Bhubaneswar, under Department of Biotechnology and National Institute of Science Education and Research (NISER), Bhubaneswar, under Department of Atomic Energy (DAE), Government of India. We wish to acknowledge the University Grant Commission (UGC), New Delhi, India for the fellowship to PM during this study. The funders had no role in study design, data collection and analysis, decision to publish, or preparation of the manuscript.

**Competing interests:** The authors have declared that no competing interests exist.

## Author summary

Chikungunya virus has been a dreaded disease from the first time it occurred in 1952 Tanzania. Since then it has been affecting the different parts of the world at different time periods in large scale. It is typically transmitted to humans by bites of *Aedes aegypti* and *Aedes albopictus* mosquitoes. Although, studies have been undertaken to combat its prevalence still there are no effective strategies like vaccines or antivirals against it. Therefore it is essential to understand the virus and host interaction to overcome this hurdle. In this study two host factors MK2 and MK3 have been taken into consideration to see how they affect the multiplication of the virus. The *in vitro* and *in vivo* experiments conducted demonstrated that inhibition of MK2 and MK3 not only restricted viral release but also decreased the disease score and allowed better survivability. Therefore, MK2 and MK3 could be considered as the key targets in the anti CHIKV approach.

## Introduction

The Chikungunya virus (CHIKV) is an insect-borne virus belonging to the genus *Alphavirus* and family *Togaviridae* and transmitted to humans by *Aedes* mosquitoes[1]. Three CHIKV genotypes, namely West African, East Central South African and Asian have been identified. The incubation period ranges from two to five days following which symptoms such as fever (up to 40˚C), petechial or maculopapular rash of the trunk and arthralgia affecting multiple joints develop[2–4].

CHIKV is a spherical (60-70nm diameter) enveloped, positive sense single-stranded RNA (~12Kb) virus [5–7]. Its genomic organization is 5′-cap-nsP1-nsP2-nsP3-nsP4-(junction region)-C-E3-E2-6K-E1-3′[8]. The non-structural proteins (nsP1-4) are primarily involved in virus replication, while structural proteins C, E3, E2, 6K and E1 are responsible for packaging and producing new virions.

In India, CHIKV infection has re-emerged with the outbreak of 2005–08 affecting approximately 1.3 million people in 13 different states [9]. The clinical manifestations during these outbreaks were found to be more severe leading to the speculation that either a more virulent or an efficiently transmitted variant of this virus might have emerged [10].

CHIKV, among most other viruses across families, interacts with a number of cellular proteins and consequently metabolic pathways to aid its survival in the host [11–17]. Several facets of CHIKV pertaining to strategies required for ecological success, replication, host interaction and genetic evolution are yet to be fully explored and are constantly evolving. This spurs the need to identify important host pathways that can be targeted for developing antiviral therapies against the virus.

Alternatively, host factors involved in viral replication may also be targeted. Previous studies have shown compounds targeting furin, protein kinases, and Hsp90, are inhibiting CHIKV replication *in vitro* [18–20]. However, further validation through *in vivo* experiments and preclinical studies need to be performed prior to developing effective antivirals. Literature also displays an array of different viruses like Zika virus, HBV, DENV, Influenza virus, HIV and HCMV for utilizing MAPKs like P38MAPK and its substrates for establishing persistent infection thereby ensuring their survival in the system for efficient progeny formation [21–27]. The MAPK-activated protein kinases MAPKAPK2 (MK2) and MAPKAPK3 (MK3) form a pair of structurally and functionally closely related enzymes and are one of the substrates of p38MAPK. Although the expression level and activity of MK2 is always significantly higher

than that of MK3, the substrate spectrum of both enzymes is indistinguishable and covers proteins involved in cytokines production, endocytosis, reorganization of the cytoskeleton, cell migration, cell cycle control, chromatin remodelling and transcriptional regulation. Both enzymes are promising targets for the development of small molecule inhibitors which can be used in anti-inflammatory therapy[28]. Moreover, it was reported earlier by our group that P38MAPK is crucial for CHIKV infection [29]. Besides, our group has formerly reported an Indian outbreak strain IS, to exhibit a faster replication rate than the CHIKV prototype strain, PS *in vitro* [30]. Hence, the present study identifies host genes which are modulated differentially during IS and PS-CHIKV infection in mammalian system and explores the involvement of downstream host factors of P38MAPK pathway during virus infection using both *in vitro* and *in vivo* conditions through inhibitor studies.

## Materials and methods

### Ethics statement

The mice related experiments were performed as per CPCSEA guidelines and were approved by the Institutional Animal Ethics Committee (IAEC) committee.

### Cells, viruses, antibodies, inhibitors

Vero cells (African green monkey kidney cells), CHIKV strains, prototype strain, PS (**Accession no: AF369024.2**) and novel Indian ECSA strain, IS (**Accession no: EF210157.2**) and E2 Monoclonal antibody were gifted by Dr. M. M. Parida, DRDE, Gwalior, India. The HEK 293T cell line (Human embryonic kidney cells) was gifted by Dr. Rupesh Dash, Institute of Life Sciences, Bhubaneswar, India. Cells were maintained in Dulbecco's Modified Eagle's medium (DMEM; PAN Biotech, Germany) supplemented with 5% Fetal Bovine Serum (FBS; PAN Biotech), Gentamicin and Penicillin-Streptomycin (Sigma, USA). The anti-nsP2 monoclonal antibody used in the experiments was developed by us [31]. Cofilin monoclonal antibody was purchased from Cell Signaling Technologies (Cell Signaling Inc, USA). The pMK2 polyclonal antibody and MK3 monoclonal antibody were purchased from Santacruz Biotechnology (USA). The p-Cofilin antibody and GAPDH antibody were procured from Sigma Aldrich (USA) and Abgenex India Pvt. Ltd. (India) respectively. The anti-mouse and anti-rabbit HRP-conjugated secondary antibodies were purchased from Promega (USA). Alexa Fluor 488 and Alexa Fluor 594 antibodies were purchased from Invitrogen (USA). The MK2a inhibitor, CMPD1 was purchased from Calbiochem (Germany).

### Virus infection

The Vero cells were infected with PS/IS strains of CHIKV respectively according to the experimental requirements as reported earlier [30]. Thereafter, CHIKV infected cells were incubated for 15–18 hours post infection (hpi) following which cells and supernatants were harvested from mock, infected and drug treated samples for downstream processing.

### RNA isolation and Microarray hybridization

In the present study, the global gene expression analyses were carried out using the Agilent Rhesus GeneChip ST arrays. Sample preparation was performed according to the manufacturer's instruction (Agilent, USA). Briefly, RNA was extracted from mock and virus infected Vero cells using the RNeasy mini kit (Qiagen, Germany). Next, RNA quality was assessed by Agilent Bioanalyzer and cDNA was prepared using oligo dT primer incorporating a T7 promoter. The amplified, biotinylated and fragmented sense-strand DNA targets were generated from the

extracted RNA and hybridized to the gene chip containing over 22,500 probe sets at 65˚C for 17h at 10 rpm. After hybridization, the chips were stained, washed and scanned using a Gene Chip Array scanner.[32]

## Microarray analysis

Raw data sets were extracted after scanning the TIFF files. These raw data sets were analyzed separately using the GeneSpring GX12.0 software (Agilent Technologies, USA) followed by differential gene expression and cluster analysis. Differential gene expression analyses were performed by using standard fold change cut off > = 2.0 and > = 10.0 against IS (8hpi) vs Mock (8hpi), PS (8hpi) Vs Mock (8hpi), PS (18hpi) Vs Mock (18hpi), IS (8hpi) vs PS (8hpi) and IS (8hpi) vs PS (18hpi). The hierarchical clustering was performed using the Genesis software[33]. Functional annotation of differentially expressed genes was carried out using the PANTHER gene ontology analysis software [34].

## RNA extraction and qRT-PCR

Equal volumes of serum isolated from all groups of mice samples were taken for viral RNA isolation using the QiaAmpViral RNA isolation kit (Qiagen, USA) as per the manufacturer's instructions. RT reaction was performed with 1 µg RNA using the First Strand cDNA Synthesis kit (Fermentas, USA) as per manufacturer's instructions. Equal volume of cDNA was used for PCR amplification of E1 gene of CHIKV using specific primers [35]. The viral copy number was estimated from the Ct values obtained for CHIKV E1 gene from each sample.

## siRNA transfection

Monolayers of HEK 293T cells with 70% confluency ($1\times10^6$ cells/well) in 6-well plates were transfected separately or in combination with 60pmols of siRNA corresponding to MK2 mRNA sequence [(5'-3') CCAUCACCGAGUUUAUGAAdTdT] and MK3 mRNA sequence [(5'-3') GAGAAGCUGCAGAGAUAAUdTdT] or with siRNA negative control. Transfection was performed using Lipofectamine-2000 (Invitrogen, USA) according to the manufacturer instructions. In brief, HEK cells were transfected using Lipofectamine 2000 according to different siRNA quantity in Opti-MEM medium (Thermo scientific, USA). The transfected cells were infected with either CHIKV strains PS or IS with MOI 0.1 at 24 hours post transfection (hpt). Eighteen hours post infection, the cells were harvested to measure the nsP2 and MK2/3 protein levels by Western blot analysis.

## SDS-PAGE and Western blot analysis

Protein expression was examined by Western blot analysis as described earlier [30,36]. CHIKV nsP2 and E2 proteins were detected with monoclonal antibodies [37] and re-probed with GAPDH antibody to confirm the equal loading of samples. The pMK2, MK3, Cofilin and pCofilin antibodies were used as recommended by the manufacturer. The Western blots were scanned using the Quantity One Software (Bio Rad, USA).

## Plaque assay

The CHIKV-infected cell culture supernatants were collected at 18 hpi and subjected to plaque assay according to the procedure mentioned earlier [38].

## Immunofluorescence staining

Immunofluorescence staining was carried out using the procedures described earlier [31]. Vero cells were grown on glass coverslips placed in 35mm dishes and infected with CHIKV (MOI 0.1) as described above. At 18 hpi, coverslips were stained with primary antibodies followed by staining with secondary antibody (AF 594-conjugated anti-mouse antibody) for 45 mins. The phalloidin staining was carried out using the Cytopainter F actin labeling kit as per manufacturer's protocol (Abcam, UK). The coverslips were stained with DAPI for 90 sec and mounted with 15–20 μl Antifade (Invitrogen, USA) to reduce photo-bleaching. Fluorescence microscopic images were acquired using the Leica TCS SP5 confocal microscope (Leica Microsystems, Germany) with 63X objective and analyzed using the Leica Application Suite Advanced Fluorescence (LASAF) V.1.8.1 software.

## Immunohistochemistry analysis

For histopathological examinations, tissue samples were dehydrated, embedded in paraffin wax, and thereafter serial paraffin sections (5μm) were obtained [39]. Briefly, the sections were immersed in two consecutive xylene washes for de-paraffinization and were subsequently hydrated with five consecutive ethanol washes in descending order of concentration: 100%, 90%, 70%, and deionized water. The paraffin sections were then stained with hematoxylin-eosin (H&E), and histopathological changes were visualized using a light microscope (Zeiss Vert.A1, Germany).

## Cellular cytotoxicity assay

Cellular cytotoxicity assay was performed as described earlier [40]. Vero cells were seeded onto 96-well plates at a density of 3000 cells/well, treated with different concentrations of CMPD1 for 24 hrs at 37˚C with 5% $CO_2$. DMSO-treated samples served as control. After incubation, 10μl of MTT reagent (Sigma Aldrich, USA) was added to the wells followed by incubation at 37˚C for 3hrs and processed further. Absorbance of the suspension was measured at 570nm using ELISA plate reader (BioRad, USA). Cellular cytotoxicity was determined in duplicates and each experiment was repeated thrice independently.

## CMPD1 treatment

Vero cells with 90% confluency were grown in 35mm or 60mm cell culture dishes (according to the experimental requirements) and infected with PS or IS strains of CHIKV as described above at MOI 0.1. After infection, cells were treated with either DMSO or different concentrations of CMPD1 as per the protocol mentioned earlier [41]. The cells were observed for detection of cytopathic effect (CPE) under 10X objective of bright field microscope. Infected cells and supernatants were then collected at 15-18hpi depending on the experiment.

## Time of addition experiment

Vero cells were infected with CHIKV as described above and CMPD1 (50μM) was added at 1hr interval up to 11hrs to the infected cells in different dishes. Thereafter, cell culture supernatants of all the samples were harvested at 15hpi and plaque assay was carried out for estimating viral titer.

## Transmission electron Microscopy (TEM) imaging

Ultra-structure of CHIKV particles were analyzed by TEM as mentioned before [42] In brief, mock, CHIKV-infected and drug treated cells were harvested at 22 hpi and fixed in 2%

glutaraldehyde in 0.1 M cacodylate buffer at 4˚C for overnight, followed by washing and post-fixation with 1% osmium tetroxide at RT. Next, the fixed cells were again washed, dehydrated serially in ethanol, placed in a mixture of (11) propylene oxide/Epon resin and left overnight in pure resin for impregnation of the samples prior to embedding in Epon resin, which was allowed to polymerize for 48 hours at 60˚C [43]. The hardened blocks were then cut into ultra-thin sections (70 nm thick) using a Leica EM UC7 ultramicrotome. The sections were collected on copper-grids, double-stained with 2% uranyl acetate and 1% lead citrate to enhance contrast, washed, dried and stored for subsequent TEM imaging.

## CHIKV infection in mice

Around 10–14 days old male C57BL/6 mice (n = 5) were injected subcutaneously with $1x10^6$ particles of IS in DMEM. At 3hpi, mice were fed with CMPD1 at a concentration of 5mg/kg of body weight and continually fed at every 24hr-interval up to3 days. All mice were sacrificed on the fourth day; blood samples were harvested from mock, infected and drug-treated samples and used for downstream processing. For survival curve analysis, CHIKV-infected mice were fed with CMPD1 and observed every day, for CHIKV-induced disease manifestations up to 8 days post infection (dpi). All infected mice were scored on a scale of 0 to 6 based on CHIKV induced disease symptoms such as(0- No symptoms, 1- lethargic, 2- ruffled fur, 3- restricted movement/limping, 4- one hind limb paralysis, 5 –both hind limb paralysis and 6- Morbid/dead).

## Proteome profiling

In order to assess the levels of different cytokines in mock, CHIKV-infected and CHIKV-infected+drug treated mice samples, proteome profiling was performed using the Mouse XL cytokine array kit (R & D systems, USA) as per manufacturer's instructions. The array blots were incubated with serum samples at 4˚C overnight on a gel rocker, followed by incubation with HRP-conjugated secondary antibody. Blots were developed using the chemiluminescent HRP substrate and scanned by the Image Lab software (Bio-Rad, USA). The relative differences in expression patterns of selected cytokines among the different groups of samples were assessed using the GraphPad Prism 8.0 software.

## Bioavailability Prediction

The bioavailability of CMPD1 was predicted through SWISS ADME tool available in the website (www.swissadme.ch). The SMILE structure of CMPD1 was submitted to the tool for analysis and prediction.

## Statistical analysis

Statistical analysis of the experimental data was performed by using the GraphPad Prism 8.0 Software and presented as mean±SD of three independent experiments. The One-way ANOVA with Dunnet post-hoc test was used to compare the differences between the groups. In all the tests, $p$ value < 0.05 was considered to be statistically significant.

# Results

## Differential host gene expressions for PS and IS strains of CHIKV in Vero cells

Earlier, it was observed that CPE developed by IS was more prominent at 8 hpi as compared to PS which showed similar CPE around 18 hpi [30]. To understand the host gene expression

profiles for the two CHIKV strains, Vero cells were harvested at 8 hpi for IS and 18 hpi for PS for microarray analysis to compare the host factor modulation in the same phase of replication between the strains. Microarray data revealed the differential expression of 20227 genes, of which 12221 genes were differentially expressed after applying fold change cut off ≥2.0. Further, 684 genes from the 12221 were differentially expressed with fold change ≥10.0. The cluster analysis of differentially expressed genes was carried out using the GENESPRING GX 12.0 software, as shown in Fig 1A. Annotation of the total genes into different protein classes was carried out using the Panther software. It was observed that majority of the genes belonged to the nucleic acid binding molecules, signaling molecules, transcription factors among others, as represented in Fig 1B. A pie-chart was constructed using the Panther software to annotate these genes into different biological processes, and it was observed that majority of the modulated genes belonged to the pathways involved in different cellular processes (Fig 1C). Moreover, 720 genes were differently regulated by IS alone as depicted by the Venn diagram constructed through the Gene Venn software (Fig 1D and 1E). Out of these 720 genes, few selected genes were functionally annotated into different host cellular pathways as shown as Table A in S1 Data. MK3 was present among the 720 genes that were differently expressed in IS infected cells at 8 hpi in comparison to PS (8 and 18 hpi). Thus, the importance of MK3 and its isozyme partner MK2 was deliberated during CHIKV infection in this study. Together, the data indicate that CHIKV utilizes different host cell pathways for efficient replication inside the host cell and there are differential host gene expression patterns for various strains of CHIKV.

## MK2 and MK3 gene silencing abrogates CHIKV progeny release without affecting viral protein synthesis

To elucidate the importance of MK2 and MK3 in CHIKV infection, gene silencing through siRNA approach was employed. Since the transfection efficiency of Vero cell is poor, HEK 293T cell line was used for this experiment. HEK293T cells were transfected with 60 pmol of MK2 and/or MK3 siRNAs and incubated for 24 hrs at 37°C.

Next, the siRNA transfected cells were infected with CHIKV [(PS/IS), MOI 0.1] and cells as well as supernatants were harvested at 18 hpi for further analysis. No remarkable change in nsP2 expression was observed after genetic knock down of either MK2 or MK3. Surprisingly, the expression of CHIKV-nsP2 was increased marginally when both MK2 and MK3 were silenced together as compared to control as shown in Fig 2A and 2B (left and right panels respectively).

Next, plaque assay was performed to assess the effect of MK2 and/or MK3 down-regulation in viral progeny formation. Interestingly, it was observed that the viral titers were reduced by 58% for PS strain and 53% for IS strain as shown in Fig 2C and 2D. Therefore, it can be suggested that MK2 and MK3 altogether affects CHIKV progeny release without affecting viral protein synthesis.

## CMPD1, an MK2a inhibitor abrogates CHIKV infection *in vitro*

The MAPK-activated protein kinases MAPKAPK3 (MK3) and MAPKAPK2 (MK2) are the substrates of P38 MAPK that form a pair of structurally and functionally closely related enzymes. Being highly homologous enzymes (around 70% at the amino acid sequence), their substrate spectrums are indistinguishable [44]. MK2 expression levels usually exceeds MK3 level in cells, however, in absence of functional MK2, MK3 compensates.

To investigate the role of MK2 pathway in CHIKV infection, Vero cells were treated with a non-ATP competitive MK2 inhibitor, CMPD1, which selectively inhibits P38-mediated MK2

**(A)**

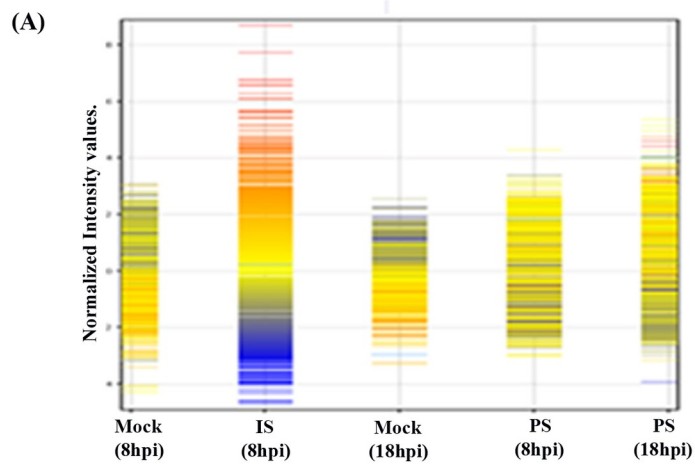

**(B)** 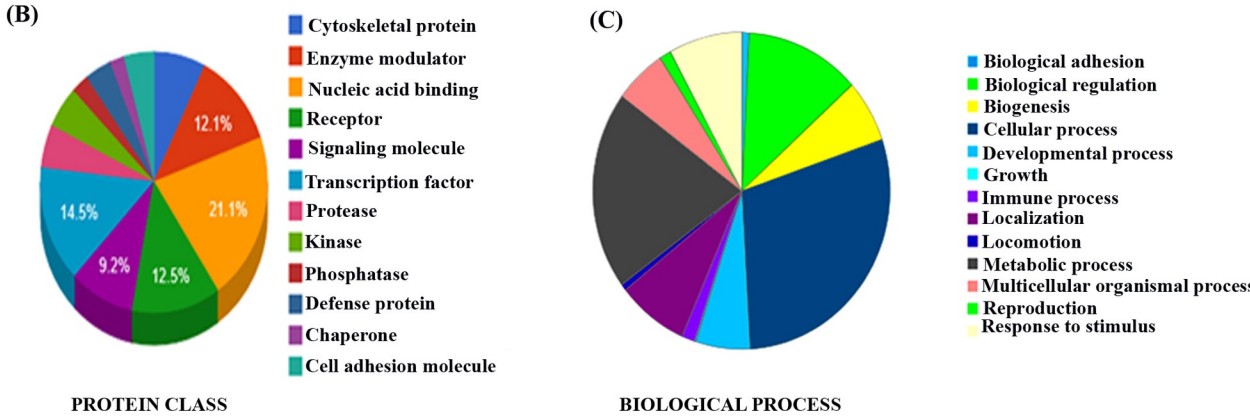 **(C)**

PROTEIN CLASS

BIOLOGICAL PROCESS

**(D)** 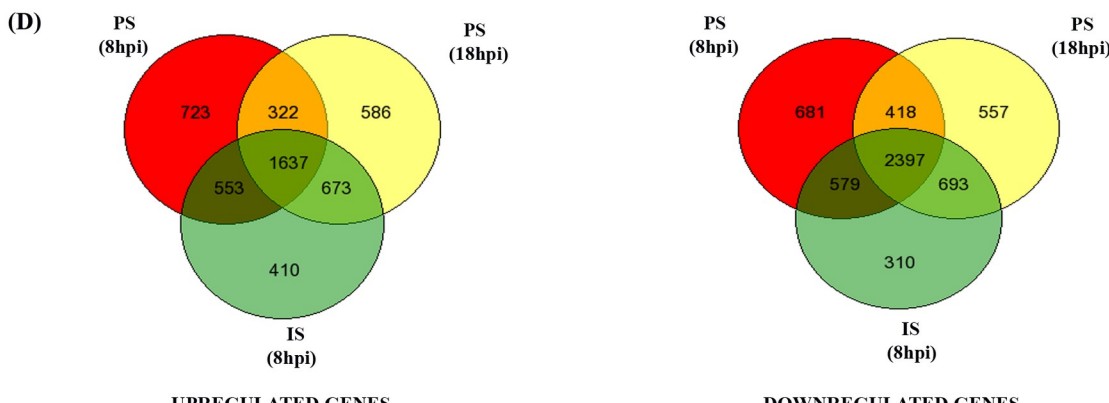

UPREGULATED GENES

DOWNREGULATED GENES

**Fig 1. Differential host gene expressions for PS and IS strains of CHIKV in Vero cells. (A)** Hierarchical clustering showing the overall expression patterns of the modulated host genes by PS/IS strains of CHIKV during infection in mammalian cells. **(B)** Pie-chart depicting the distribution of the host genes in CHIKV-infected samples into different protein classes. **(C)** Pie-chart depicting the distribution of the modulated host genes into different cellular processes. **(D and E)** Venn diagram showing both commonly and differentially regulated host genes in CHIKV (PS/IS) infected Vero cells.

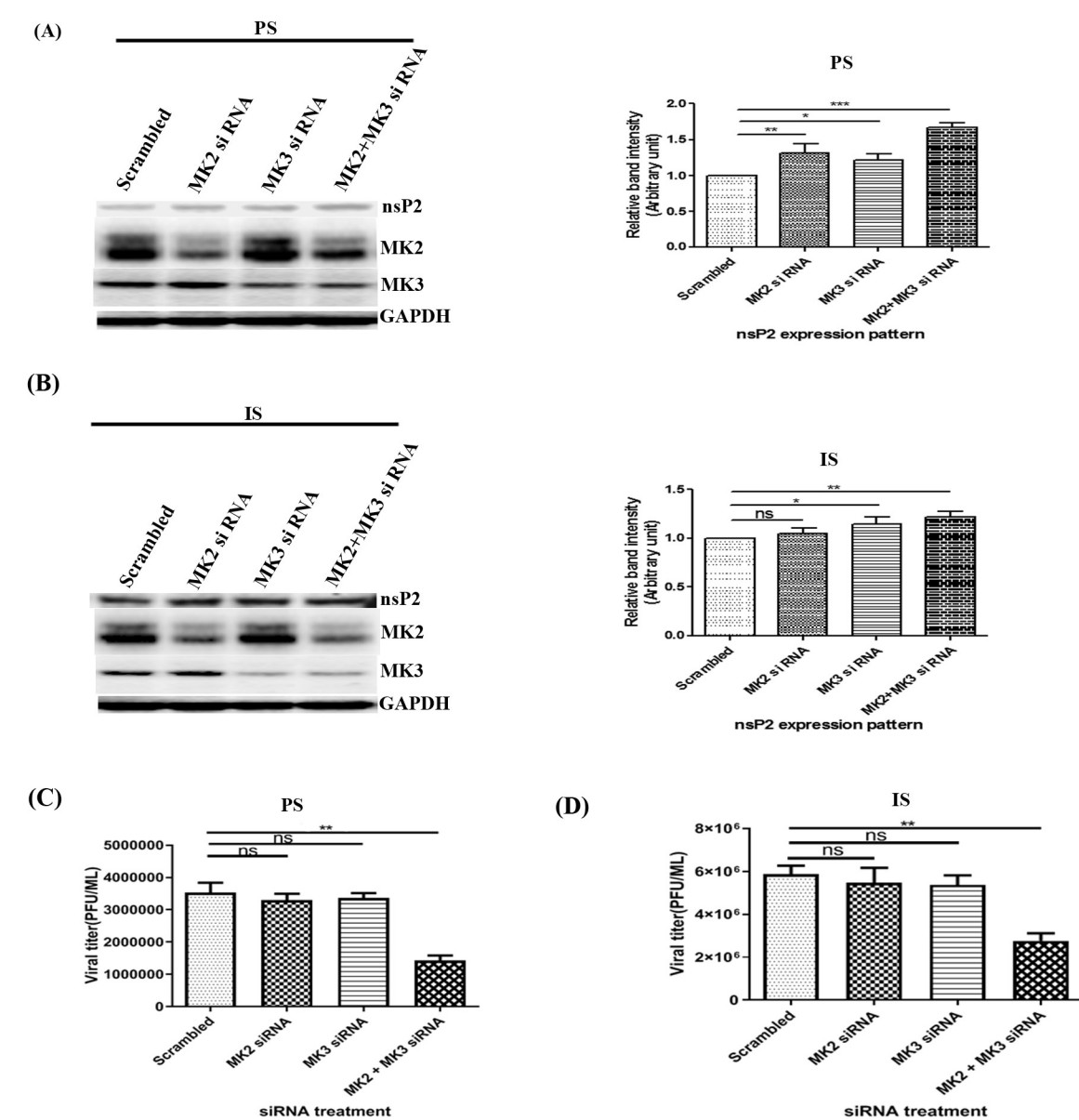

**Fig 2. MK2 and MK3 gene silencing abrogates CHIKV progeny release without affecting viral protein synthesis. (A and B)** After 24 hrs post transfection with 60 pmol of MK2/3 siRNA (either separately/in combination), cells were super-infected (PS/IS MOI 0.1) and harvested at 18 hpi. Western blot showing the expression levels of different proteins (Left panel). Bar diagrams showing relative band intensities of different proteins (Right panel). GAPDH was used as control. **(C and D)** Bar diagram showing the viral titres after siRNA treatment for PS and IS strains, (n = 3; $p<0.05$).

activation [45]. In order to determine the cytotoxicity of CMPD1, Vero cells were treated with different concentrations of the drug (25 to100μM) for 24 h and MTT assay was performed. It was observed that 98%, 95% and 85% cells were viable with 25, 50 and 100μM concentrations of the drug, respectively, as shown in Fig 3A. Next, dose kinetics assay was performed to determine the anti-CHIKV efficacy of CMPD1 by infecting Vero cells with two different strains of CHIKV with MOI 0.1 and treated with 25, 50 and 100μM concentrations of CMPD1. The cell culture supernatants were harvested at 18 hpi and plaque assay was carried out to estimate the virus titers. Around 90% decrease in virus titer was observed with higher concentrations of

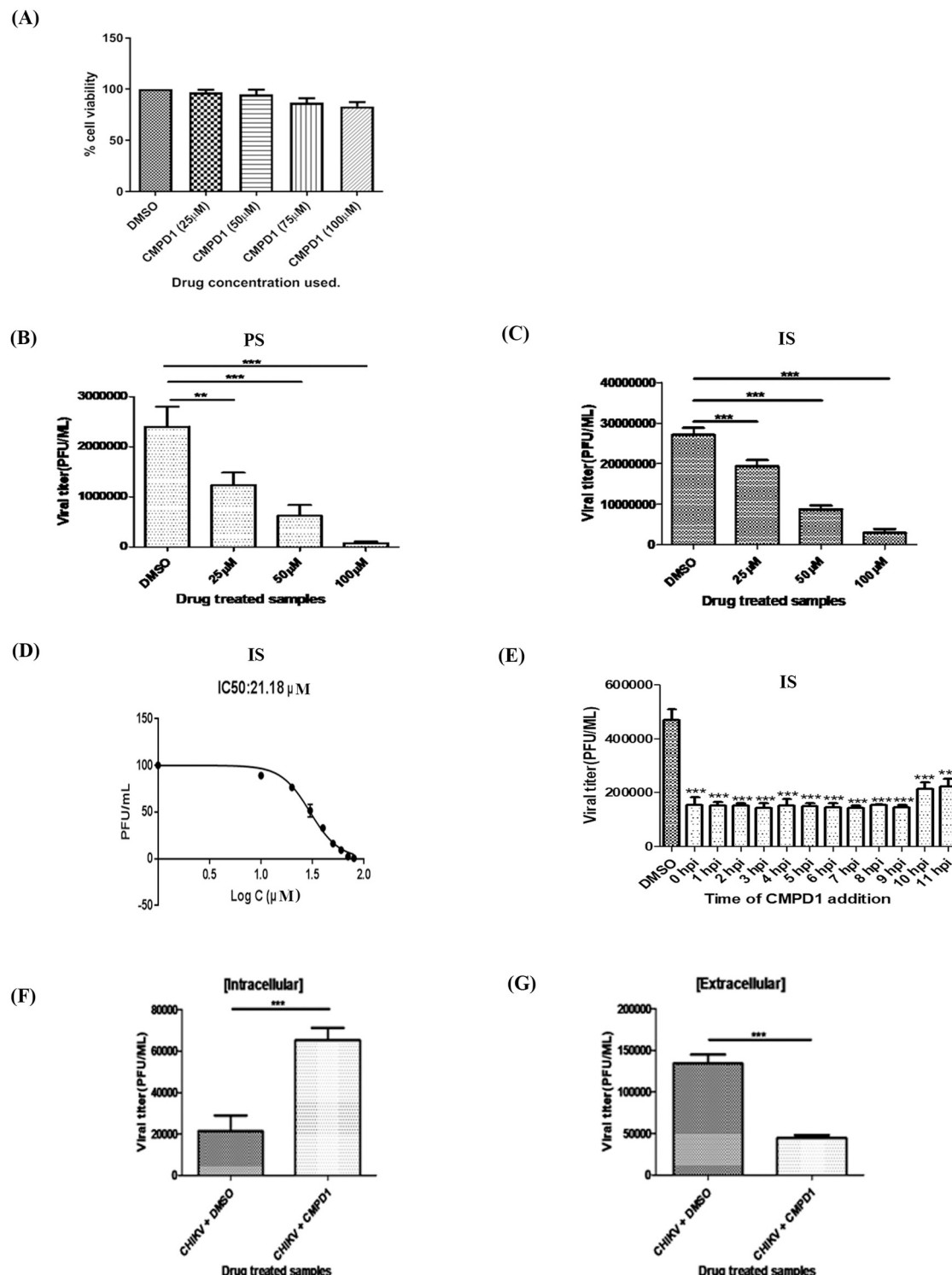

**Fig 3. CMPD1, an MK2a inhibitor abrogates CHIKV infection *in vitro*. (A)** Vero cells were treated with different concentrations of CMPD1 (25, 50, 75 and 100 μM) for 24 h and MTT assay was performed. **(B and C)** Vero cells infected with CHIKV PS/IS at MOI 0.1 and drug treated. Bar graph showing viral titers in the presence of CMPD1 (25, 50 and 100 μM). **(D)** Dose response curve showing the $IC_{50}$ of CMPD1 against CHIKV. **(E)** Bar graph showing the viral titers estimated through plaque assay from the supernatants obtained from the time of addition experiment for CMPD1 (50μM) post CHIKV infection. **(F and G)** Bar graph showing intracellular and extracellular virus titers for samples harvested at 18hpi. DMSO was used as control. All the graphs depict the values of mean ± SD (*p< 0.05*) of three independent experiments.

CMPD1 in comparison to DMSO control for both the strains (Fig 3B and 3C). Additionally, similar experiment was carried out in HEK cell line using 5 and 10μM of CMPD1 and it was found that around 80% decrease in viral titer was obtained at the higher dose of CMPD1(Fig A in S1 Data) confirming the importance of MK2 for CHIKV in human cell line. Since, effect of CMPD1 was same for both the strains, IS strain (more virulent of the two strains used in this study) was used for further experiments.

To estimate the $IC_{50}$ value of CMPD1, Vero cells were infected with CHIKV as mentioned above and different concentrations of CMPD1 (10–80μM) were added to the cells post-infection. The supernatants were harvested at 18 hpi and plaque assay was performed. The plaque numbers were converted into log 10 of PFU/mL and the $IC_{50}$ of CMPD1 was found to be 21.18μM (Fig 3D).

Next, to assess the possible mechanism of action of CMPD1 on CHIKV replication, time of addition experiment was performed. Vero cells were infected with IS strain with MOI 0.1 and 50 μM of CMPD1 was added at 1hr interval from 0–11 hpi. DMSO was used as a control. Next, the supernatants were harvested at 15 hpi and plaque assay was performed. As shown in Fig 3E, it was observed that around 55% of the infectious virus particle release was abrogated in the presence of 50 μM of CMPD1, even after the addition of the drug at 11 hpi. This indicates that CMPD1 inhibits later phase of CHIKV life cycle like packaging or release.

For that infected and drug treated supernatants were collected at 18 hpi for estimating the extracellular viral titer through plaque assay. For estimating the intracellular virus titer, cells were washed twice with 1X PBS and harvested. The pelleted cells were resuspended in fresh serum free medium and freeze-thawed thrice to release virus particles trapped inside the cells. Then the intracellular virus titer was estimated., The extracellular virus titer was around 70% less in CMPD1 treated samples in comparison to control but the intracellular virus titer was around 60% more for CMPD1 treated samples as shown in Fig 3F and 3G. In order to strengthen the above findings, TEM imaging was carried out for CHIKV infected and CMPD1 treated cells harvested at 18 hpi and it was observed that more number of virus particles are seen trapped inside the CMPD1 treated infected cells in comparison to the untreated control as shown as Fig B in S1 Data.

Taken together, the data suggest that CMPD1 did not inhibit the formation/packaging of newly synthesized virus particles inside the host cell, however, it affects the release of CHIKV viral progeny from the host cell.

## CMPD1 blocks the actin polymerization process modulated by CHIKV for its progeny release

It is well known that both the isozymes, MK2 and MK3 are exclusively phosphorylated by P38 MAPK [46]. It is also known that LIM kinase 1 (LIMK1), a downstream substrate of MK2 induces actin polymerization by phosphorylating and inactivating cofilin, an actin-depolymerizing factor [47,48]. Therefore, to understand the effect of CMPD1 on downstream substrates of MK2, the cells were infected with IS at MOI 0.1 and treated with 50 μM CMPD1. Infected cells were observed for the development of CPE at 18 hpi and clear reduction in CPE was observed after CMPD1 treatment (Fig 4A). The cells were harvested at 18 hpi and cell lysates were processed for Western blot analysis. It was noticed that the levels of pMK2 and MK3 were downregulated after drug treatment with no change in CHIKV nsP2 expression as shown in Fig 4B and 4C. Similarly, the expression of Cofilin and p-Cofilin were decreased in the presence of CMPD1. The expression of pMK3 could not be tested due to unavailability of a commercial antibody. Altogether, the data suggest that MK2 phosphorylation plays an important role in viral progeny release by modulating the actin polymerization process.

**(A)**

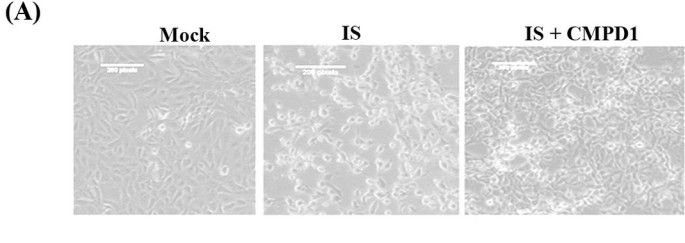

**(B)**

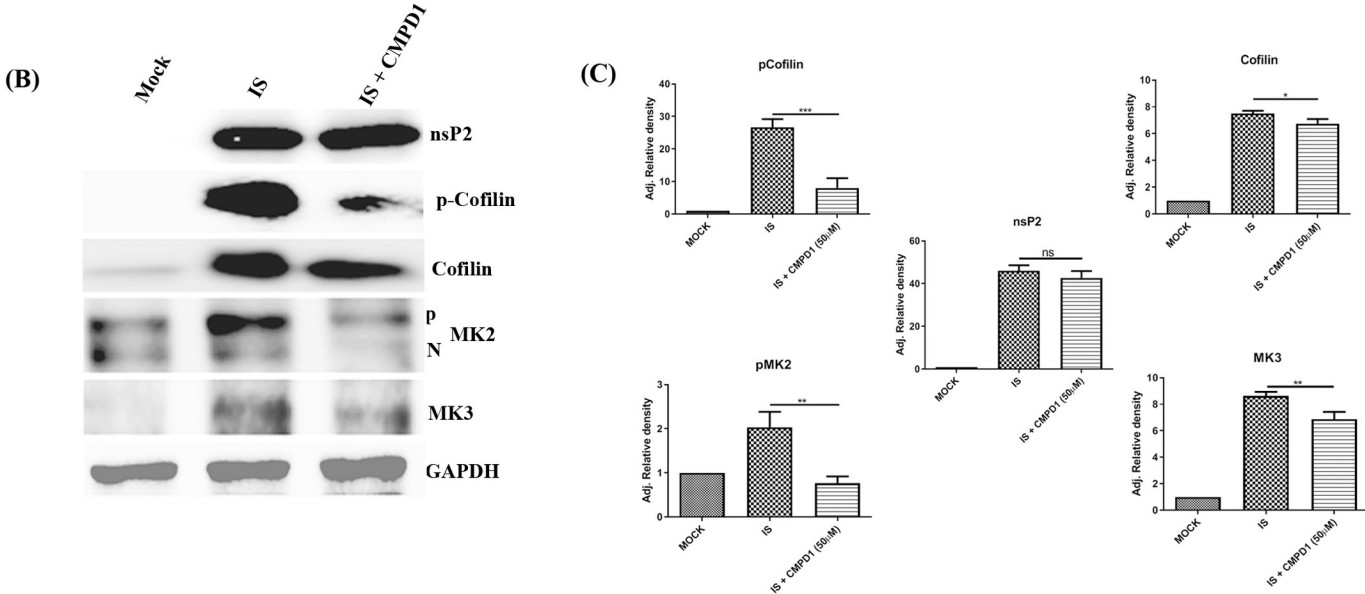

**(C)**

**(D)**

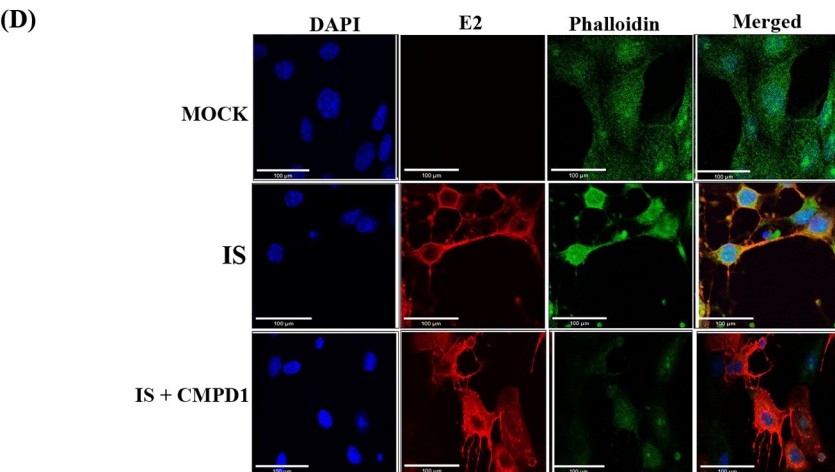

**Fig 4. CMPD1 blocks the actin polymerization process modulated by CHIKV for its progeny release.** Vero cells were infected with the IS strain (0.1 MOI), 50μM of CMPD1 was added to the cells and incubated for 18 hpi. **(A)** Bright field images (20X magnification) showing the cytopathic effect after CHIKV infection with or without CMPD1 treatment (50μM). **(B)** Western blot analysis showing the expressions of nsP2, pMK2, MK3, Cofilin and p-Cofilin proteins. GAPDH served as the loading control. **(C)** Bar graphs showing the relative fold change in viral and host proteins expression with respect to DMSO control. **(D)** Confocal microscopy images showing the levels of E2 and phalloidin during CHIKV infection.

In order to confirm the involvement of actin fibers in CHIKV progeny release, infected and drug treated cells were fixed at 18 hpi. Thereafter, phalloidin staining was carried out to stain actin fibers in cells as it has been reported that fluorescent dye-labeled phalloidin stains only the actin fibers, but not the monomers [49]. Phalloidin staining was found to be more prominent in infected cells without CMPD1 treatment and was more diffusely stained in CMPD1 treated cells. Furthermore, the expression pattern of CHIKV E2 protein was unchanged in both the samples as shown in Fig 4D. Taken together, the results depict that CHIKV utilizes the actin polymerization process for its progeny release through activation of MK2/MK3; however CMPD1 abrogates the whole process by inhibiting MK2/3 activation.

## CMPD1 inhibits CHIKV infection *in vivo*

In order to assess the bio-availability of a drug/inhibitor, computer models have been used as a valid alternative to experimental procedures for prediction of ADME (Absorption, Distribution, Metabolism and Excretion) parameters [50]. The SwissADME Web tool (www.swissadme.ch) is one such tool which enables the computation of key physicochemical, pharmacokinetic, drug- like and related parameters for one or multiple molecules [51]. It was found that CMPD1 has high GI (Gastro Intestinal) absorption with a bioavailability score of 0.55 as shown as Table B in S1 Data.

In order to assess the *in vivo* antiviral effect of CMPD1, 10–14 days old male C57BL/6 mice (n = 5 per group) were infected with the IS strain and serum as well as tissue samples were harvested as per the protocol mentioned above. Viral RNA was isolated from the pooled serum samples (from respective group) and RT-PCR was carried out to amplify E1 gene of CHIKV. It was observed that the viral copy number was reduced remarkably (90%) in CMPD1 treated CHIKV infected mice in comparison to control (Fig 5A). Next, to compare the extent of tissue inflammation, muscle tissue sections (from the site of injection) of the sacrificed mice at 4dpi were stained using Haematoxylene and Eosin and it was found that the infiltration of immune cells were less in CMPD1 treated tissue in comparison to control (Fig 5B). Furthermore, to determine the levels of different cytokines/chemokines in CMPD1 treated mice, proteome profiling was carried out with the pooled serum samples as described above. It was noticed that the expressions of few selective inflammatory cytokines/chemokines, like CXCL13, RAGE, FGF and MMP9 were significantly reduced in CMPD1 treated mice sera, (Fig 5C and 5D). Interestingly, HGF was upregulated in CMPD1 treated mice. To assess the protective action of CMPD1, survival curve analysis was performed. For that, CHIKV infected mice (5 per group) were fed with CMPD1 (5mg/kg) orally at 3hrs post CHIKV infection and then for 3 consecutive days at an interval of 24 hrs. The disease scoring was performed based on the symptoms described in the methods section and shown as Table C in S1 Data., There was 100% mortality of the untreated CHIKV infected mice after 8 days post infection (Fig 5E). In contrast, no mortality was observed for the CMPD1 treated CHIKV infected mice even after 8 days post infection. The data suggest that CMPD1 inhibits CHIKV infection *in vivo*.

## Discussion

Due to lack of therapeutics and vaccine, a number of studies have been initiated to understand the function of viral proteins and the mechanisms of virus-mediated manipulation of host machineries for successful infection [52–54]. Microarray analysis was carried out for mock and CHIKV-infected Vero cells to determine how host proteins are modulated during CHIKV infection in mammalian cells and two genes MK2 and MK3 belonging to P38MAPK pathway were selected for further analysis.

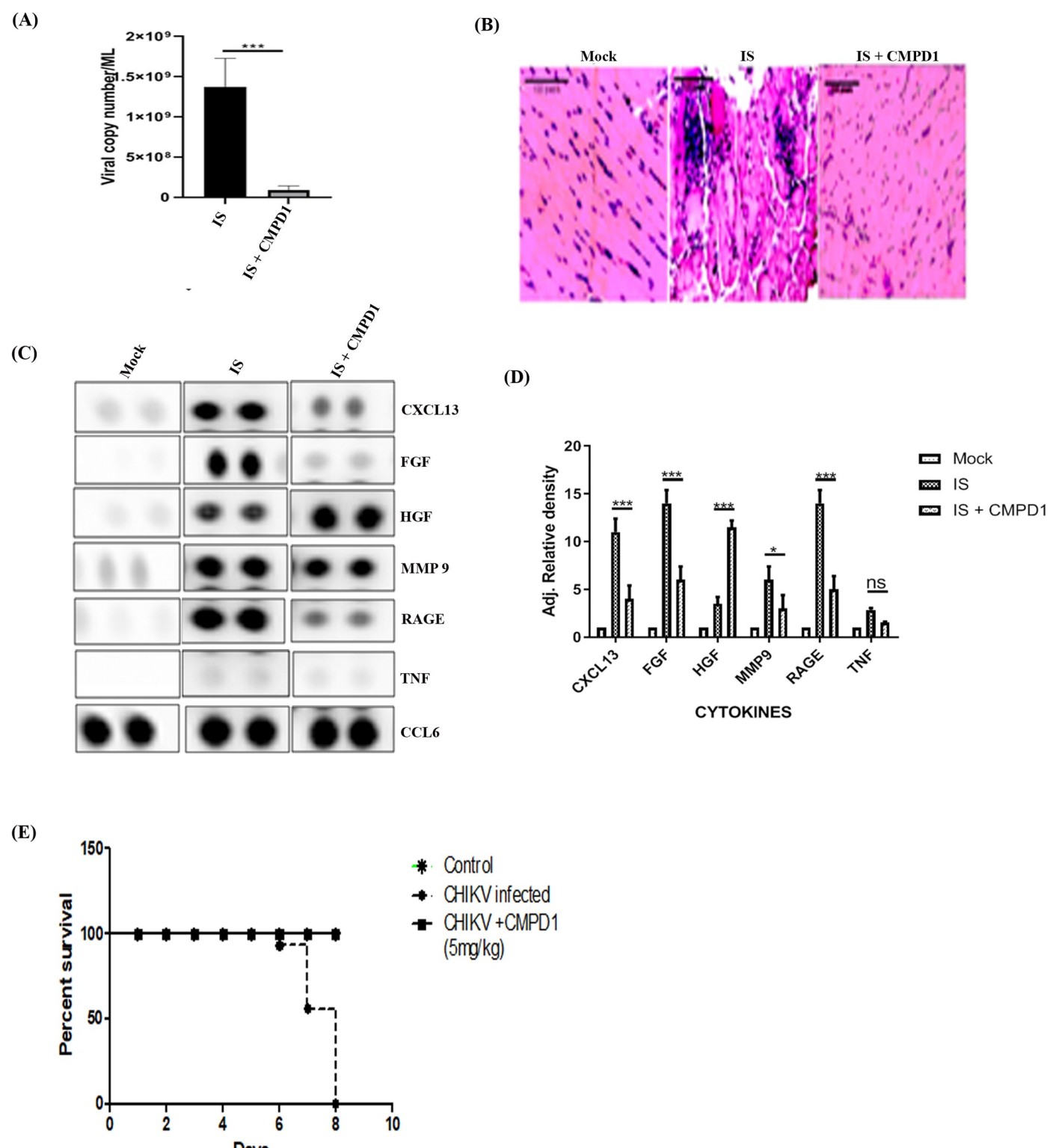

**Fig 5. CMPD1 inhibits CHIKV infection in mice. (A)** Bar graph showing the viral copy numbers in CHIKV infected and CMPD1 treated mouse serum samples. (B) H and E staining of mouse tissue samples with CHIKV infection and in presence/absence of CMPD1 **(C)** Array blot showing the expression of different cytokines after CHIKV infection in presence and absence of CMPD1. **(D)** Bar graph showing the relative band intensities of selected cytokines in mock, CHIKV infected and CMPD1 treated samples**. (E)** Survival curve showing the effect of CMPD1 in CHIKV infected mice.

Gene silencing of MK2 and MK3 abrogated around 58% CHIKV progeny release from the host cell and a MK2 activation inhibitor (CMPD1) treatment demonstrated 68% inhibition of viral infection suggesting a major role of MAPKAPKs during late CHIKV infection *in vitro*. Further, it was observed that the inhibition in viral infection is primarily due to the abrogation of lamellipodium formation through modulation of factors involved in the actin cytoskeleton remodeling pathway. Moreover, CHIKV-infected C57BL/6 mice demonstrated reduction in the viral copy number, lessened disease score and better survivability after CMPD1 treatment. In addition, reduction in expression of key pro-inflammatory mediators such as CXCL13, RAGE, FGF, MMP9 and increase in HGF (a CHIKV infection recovery marker) was observed indicating the effectiveness of the drug against CHIKV.

Role of MK2 and MK3 have been implicated in several other viruses. In DENV, it was found that SB203580 (a P38MAPK inhibitor) treatment significantly reduced the phosphorylation of MAPKAPK2 and other substrates such as HSP27 and ATF2 in mice. In the case of Murine Cytomegalovirus (MCMV), MK2 has been reported to regulate cytokine responses towards acute infection, via IFNARI-mediated pathways during infection [55]. For Kaposi Sarcoma Herpes Virus (KSHV), it was observed that the viral Kaposin B (KapB) protein binds and activates MK2, thereby selectively blocking decay of AU-rich mRNAs (ARE-mRNAs) that encode pro-inflammatory cytokines and angiogenic factors during latent KSHV infections [56]. Furthermore, it was noticed that during Rous Sarcoma Virus (RSV) infection, pP38 is sequestered inside cytoplasmic inclusion bodies (IBs) resulting in substantial reduction in accumulation of MK2 and suppressing cellular responses to virus infection. Additionally, CMPD1 treatment reduced viral protein expression suggesting the importance of pMK2 in RSV protein translation [41]. In case of Influenza A, it was observed that MK2 and MK3 are activated on virus infection enabling the virus to escape the antiviral action of PKR [57]. Recently, it has been shown that CCR5-tropic HIV induces significant reprogramming of host CD4+ T cell protein production and induces MK2 expression upon viral binding to the cell surface that are critical for HIV replication [58]. However, reports pertaining to the involvement of MK2 and MK3 in alphavirus infection are not available. Hence, this investigation is one of the first to highlight the importance of MK2 and MK3 in CHIKV.

According to the results, it can be suggested that both MK2 and MK3 play important roles in CHIKV progeny release during CHIKV infection. After CHIKV infection, MK2 is phosphorylated which in turn phosphorylates LIMK1. The LIMK1 then inactivates Cofilin by phosphorylating it. This results in accumulation of more p-Cofilin inside the cell. As a result, Cofilin is unable to cleave the actin filaments into monomers. This leads to polymerization of actin filaments and subsequent lamellipodia formation which results in effective CHIKV progeny release (Fig 6A). However, CMPD1 treatment abrogates MK2/3 phosphorylation as a result of which LIMK is not able to inactivate Cofilin. Active Cofilin then cleaves actin polymers to monomers, thereby preventing lamellipodium formation and subsequent viral progeny release (Fig 6B). Furthermore, *in vivo* studies demonstrate that CMPD1 treated mice do not develop complications post CHIKV infection. This can be speculated by the reduction in the expression of some virus induced inflammatory chemokines and cytokines like CXCL13, RAGE and FGF in CMPD1 treated mice sera [59–61]. Additionally, the expression of MMP9, a host factor involved in the degradation of extracellular matrix thereby promoting viral spread to neighbouring tissues was also reduced in drug-treated samples indicating abrogation of viral transmission during CMPD1 treatment[62]. In contrast, the expression of HGF (a known marker for CHIKV recovery during acute infection) was upregulated during CMPD1 treatment thereby showing the effectiveness of CMPD1 against CHIKV in mice [63]. Nevertheless, it would be interesting to understand the detailed mechanism and role of these factors during CHIKV infection in future.

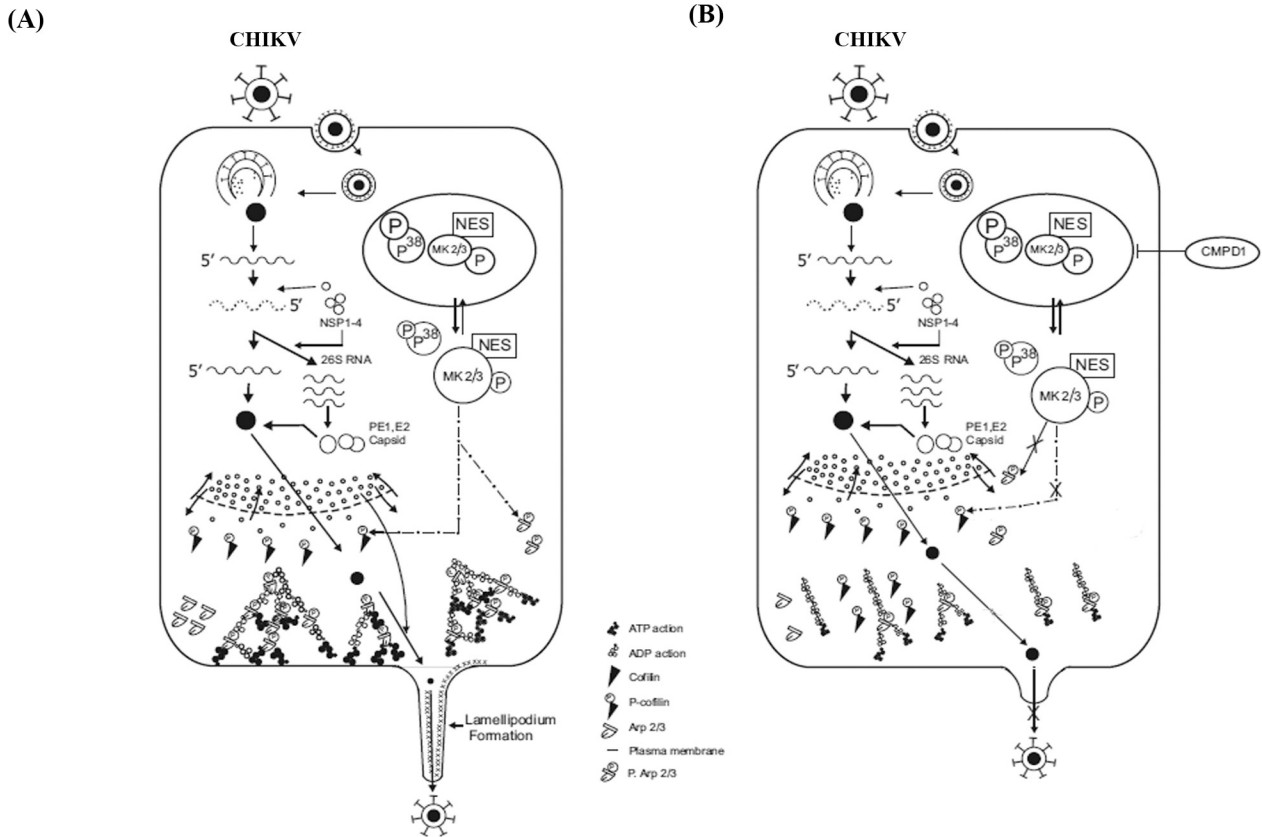

**Fig 6. Proposed model for CHIKV infection. (A)** During CHIKV infection, MK2/3 gets phosphorylated by P38 MAPK thereby exposing the Nuclear Export Signal (NES) of MK2. The phosphorylated forms of MK2/MK3 translocate to the cytoplasm and help in inactivating Cofilin through phosphorylation via LIMK-1 thereby promoting actin polymerization and lamellipodium formation. **(B)** Addition of CMPD1 blocks the phosphorylation of MK2, thereby blocking Cofilin phosphorylation and eventually inhibiting lamellipodium formation and CHIKV progeny release.

Thus, the current study highlights the importance of MK2/3 (substrates of the p38MAPK pathway) as novel host factors involved in CHIKV infection and CMPD1 can be pursued as a potential lead for developing anti-CHIKV molecule to modulate disease manifestations.

## Supporting information

**S1 Data.** Table A. Differently modulated host genes for CHIKV-IS classified in to different metabolic pathways. Table B. Bioavailability prediction of CMPD1 through the SWISSADME web tool. Table C. Disease scoring of CHIKV infected and drug treated mice. Fig A. Effect of CMPD1 on CHIKV viral titer in HEK 293T cells. Fig B. TEM image showing CHIKV particles trapped inside Vero cells during CMPD1 treatment.
(DOCX)

## Acknowledgments

We are thankful to Dr. M.M. Parida, DRDE; Gwalior, India for kindly providing IS strain of CHIKV and Vero cell line. We are also thankful to Dr. Rupesh Dash of ILS, Bhubaneswar, India for providing the HEK 293T cell line. We are grateful to the ILS TEM facility for taking the images and thankfully acknowledge the help of Mr. Aditya Anand, Mr. Cyrus Alexander and Mr. Hiren Dodia for sample processing.

## Author Contributions

**Conceptualization:** Prabhudutta Mamidi, Subhasis Chattopadhyay, Soma Chattopadhyay.

**Data curation:** Prabhudutta Mamidi, Tapas Kumar Nayak, Abhishek Kumar, Sanchari Chatterjee, Soumyajit Ghosh, Supriya Suman Keshry, Sharad Singh.

**Formal analysis:** Prabhudutta Mamidi, Tapas Kumar Nayak, Abhishek Kumar, Ankita Datey, Subhasis Chattopadhyay.

**Funding acquisition:** Subhasis Chattopadhyay, Soma Chattopadhyay.

**Investigation:** Prabhudutta Mamidi, Tapas Kumar Nayak, Sameer Kumar, Sanchari Chatterjee, Saikat De, Ankita Datey.

**Methodology:** Prabhudutta Mamidi, Tapas Kumar Nayak, Sameer Kumar, Sanchari Chatterjee, Saikat De, Ankita Datey, Soma Chattopadhyay.

**Project administration:** Soma Chattopadhyay.

**Resources:** Subhasis Chattopadhyay, Soma Chattopadhyay.

**Supervision:** Subhasis Chattopadhyay, Soma Chattopadhyay.

**Validation:** Prabhudutta Mamidi, Subhasis Chattopadhyay, Soma Chattopadhyay.

**Visualization:** Prabhudutta Mamidi.

**Writing – original draft:** Prabhudutta Mamidi, Tapas Kumar Nayak, Sameer Kumar, Saikat De, Eshna Laha, Amrita Ray.

**Writing – review & editing:** Prabhudutta Mamidi, Eshna Laha, Amrita Ray, Subhasis Chattopadhyay, Soma Chattopadhyay.

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
