## [Decision Letter · Decision Letter 0]

14 Jun 2021

Dear Dr. Chattopadhyay,

Thank you very much for submitting your manuscript "MK2a inhibitor CMPD1 abrogates chikungunya virus infection by modulating actin remodeling pathway" for consideration at PLOS Pathogens. As with all papers reviewed by the journal, your manuscript was reviewed by members of the editorial board and by several independent reviewers. In light of the reviews (below this email), we would like to invite the resubmission of a significantly-revised version that takes into account the reviewers' comments. Particular attention should be paid to Reviewer 2's comments concerning experimental evidence of p38's mechanism of action in this study, without which acceptance will be difficult. Data presentation, including in the supplemental material, should be made more clear. 

We cannot make any decision about publication until we have seen the revised manuscript and your response to the reviewers' comments. Your revised manuscript is also likely to be sent to reviewers for further evaluation.

Sincerely,

Bryan Mounce, Ph.D.

Guest Editor

PLOS Pathogens

Marco Vignuzzi

Section Editor

PLOS Pathogens

Kasturi Haldar

Editor-in-Chief

PLOS Pathogens

orcid.org/0000-0001-5065-158X

Michael Malim

Editor-in-Chief

PLOS Pathogens

orcid.org/0000-0002-7699-2064

Reviewer's Responses to Questions

**Part I - Summary**

Reviewer #1: This is a very interesting manuscript, the strength of the manuscript is authors have done a detailed study including in animal model.

Reviewer #2: This manuscript addresses a gap in the pharmacopoeia for the management of chikungunya. The authors conducted gene expression study that identified p38 MAPK pathway to be upregulated upon CHIKV infection. The authors then focused on two isozymes, MK2 and MK3, where silencing of these genes led to reduced CHIKV titers in vitro. The authors then showed that the antiviral effects shown in vitro could be translated into antiviral efficacy in a mouse model - treatment with CMPD1, an MK2 inhibitor, led to reduced viral titers and histopathological changes in the muscle. The authors also showed that this same compound could also inhibit HSV-1 and SARS-CoV-2 infection, in vitro. They concluded that MK2 and MK3 could be targeted for the development of new antiviral treatment.

Reviewer #3: Authors studied host factors related to CHIKV infection in Vero cells. By microarray analysis, MK2/MK3 were revealed as such factors, which was further confirmed by CMPD1 treatment to infected cells and animals. This study reported MK2/MK3 as critical factors for CHIKV infection which may be a key targets for antiviral therapy in future.

**Part II – Major Issues: Key Experiments Required for Acceptance**

Reviewer #1: I am not sure how microArray data has been analysed. Author may take help from a professional bio informatics person. Author could use graph theory to analyze the micro array data. What is the rational to measure these cytokines as a proinflmmatory one? Why not IL-1beta or TNFalpha, author should explain.

Reviewer #2: There are several issues with this study.

1. The authors compared two strains of CHIKV - PS and IS. The rationale why any comparison is needed at all is unclear. More importantly, the comparison complicated downstream analyses. Specifically, the time needed to complete a single round of replication appears to be different in these two strains. Based on their previous publication (ref #21), the pfu of one strain peaked at 8 hours post infection whereas the other peaked only at 20 hours post infection. Moreover, at 20 hours post infection, the faster replicating strain showed significant decline in pfu at this time point. Moreover, more than 50% of the cells infected with the faster replicating strain have shown cytopathic effects (CPE) as early as 12 hours post infection. Consequently, the comparison of the mRNA transcript levels at 20 hours post infection (line 146) is not comparing apples with apples. One strain would have undergone 3 (possibly 4) rounds of infection whereas the other would have just completed a single round of replication. The quality of the mRNA when >50% of the cells show CPE would also be doubtful.

2. Related to point 1, the effects of MK2/MK3 silencing on PS and IS strain infection are also difficult to interpret without comparing equivalent rounds of replication.

3. The use of Vero cells without validation of the hits on a relevant human cell also complicates interpretation of the findings. This is particularly since the goal of this study is to identify a suitable target for therapeutic development. Ideally, the gene expression study should have been done in a relevant human cell line.

4. The rationale for focus on MK2 and MK3 is unclear. The data in Table S1 is inadequate. The authors should show the mean and SD of each of the genes that were listed in this table. More appropriately, the entire microarray dataset should be made available for scrutiny. Without detailed data, it would seem that the authors cherry picked MK2 and MK3 for further examination, especially since the pathways highlighted in Table S1 are not consistent with those shown in Figure 1B and C.

5. The notion that MK2 and MK3 silencing led to inhibition of "CHIKV progeny release" is not well supported by the data. The authors only measured nsP2 levels by western blot. On the other hand, CHIKV replication involves multiple steps, from virus uncoating to RNA transcription and translation, to viral protein folding, vision assembly and finally egress. The authors only showed that nsP2 could be detected and was marginally increased with MK2 and MK3 knockdown. The specific step in which virus infection/replication was inhibited by the reduction in MK2 and MK3 is entirely unclear without more detailed study. The data Figure 3F and G could be argued to support the notion that CHIKV was formed but could not egress the cell with CMPD1 treatment. However, this data is preliminary and the impact of non-MK2/MK3 dependent effects of CMPD1 on viral assembly/egress has not been excluded.

6. The in vitro experiment with CMPD1 is problematic. The authors should show the actual MTT assay data. Conventionally, any statistically significant reduction in cell viability compared to mock treated control would not be used to assess for antiviral activity, as reduction in cell viability will reduce viral titers. The data described in line 323 is thus insufficient. Moreover, any less than 90% cell viability is likely to be significantly different compared to control cells. CMPD1 at 100 uM should thus not have been used to calculate the IC50. Moreover, it is also unclear why the authors chose to no report the data with 75 uM of CMPD1 (line 322).

7. Even with the use of data from 100uM CMPD1 treatment to estimate IC50, the value of nearly 34 uM is high, especially since significant level of toxicity is likely reached at 100 uM. The potential for any clinical translation is thus questionable.

8. The p38 pathway has been implicated in the pathogenesis of several other viral infections, including those discussed in this manuscript. The authors should thus have examined if some of these reported mechanisms, including control of RIPK1 signalling that is important for inflammation and infection, as reported by Menon et al (Nat Cell Biol 2017; 19:1248-59), could explain the observed outcome with CMPD1 treatment. Without excluding such known antiviral and anti-inflammatory pathways, the mechanism of action offered by the authors in Figure 7 is premature.

Reviewer #3: Authors designed elaborate study and performed many experiments, which are described in detail. However, Materials and methods section, and Results section are too lengthy. In addition, it seems abrupt to add experiments with SHV and SARS-CoV-2. Readers may feel it as in a hit-or-miss manner, because of the finding with CHIKV. Therefore, authors should considerably shorten the manuscript as much as possible, and consider to delete the part of SHV and SARS-CoV-2, which may be published as other report.

**Part III – Minor Issues: Editorial and Data Presentation Modifications**

Reviewer #1: Discussion part is too lengthy.

Reviewer #2: Line 46. "effective" should be changed to "licensed".

Line 97. CHIKV is not conventionally described as small.

Line 292. What is "antagonically"?

Reviewer #3: Authors identified MK2/MK3 as host factors, and their inhibitor CMPD1 was used for further analysis. However, there is no explanation of them in Abstract, and names of them did not appear in Introduction. Readers who are not familiar with them may not understand this study, which is not appropriate as scientific article. If word limit is permitted, any brief explanation of MK2/3 and CMPD1 should be added to Abstract and Introduction. For example, MK2 as MAPK-activated protein kinase 2; MK2 is a checkpoint kinase involved in the DNA damage response, etc. Line 435-436, "MK2 and MK3 belonging to P38MAPK pathway" needs some more annotations.

PLOS authors have the option to publish the peer review history of their article (what does this mean?). If published, this will include your full peer review and any attached files.

Reviewer #1: No

Reviewer #2: No

Reviewer #3: No
---

## [Editor Report · Decision Letter 1]

15 Oct 2021

Dear Dr. Chattopadhyay,

We are pleased to inform you that your manuscript 'MK2a inhibitor CMPD1 abrogates chikungunya virus infection by modulating actin remodeling pathway' has been provisionally accepted for publication in PLOS Pathogens.

Best regards,

Bryan Mounce, Ph.D.

Guest Editor

PLOS Pathogens

Marco Vignuzzi

Section Editor

PLOS Pathogens

Kasturi Haldar

Editor-in-Chief

PLOS Pathogens

orcid.org/0000-0001-5065-158X

Michael Malim

Editor-in-Chief

PLOS Pathogens

orcid.org/0000-0002-7699-2064
---

## [Editor Report · Acceptance letter]

29 Oct 2021

Dear Dr. Chattopadhyay,

We are delighted to inform you that your manuscript, "MK2a inhibitor CMPD1 abrogates chikungunya virus infection by modulating actin remodeling pathway," has been formally accepted for publication in PLOS Pathogens.

Best regards,

Kasturi Haldar

Editor-in-Chief

PLOS Pathogens

orcid.org/0000-0001-5065-158X

Michael Malim

Editor-in-Chief

PLOS Pathogens

orcid.org/0000-0002-7699-2064